# Knowledge and Perception of and Attitude toward a Premarital Screening Program in Qatar: A Cross-Sectional Study

**DOI:** 10.3390/ijerph19074418

**Published:** 2022-04-06

**Authors:** Mashael Al-Shafai, Aisha Al-Romaihi, Noora Al-Hajri, Nazmul Islam, Karam Adawi

**Affiliations:** 1Department of Biomedical Sciences, College of Health Science, QU Health, Qatar University, Doha 2713, Qatar; aa1402887@student.qu.edu.qa (A.A.-R.); na1403240@student.qu.edu.qa (N.A.-H.); 2Department of Public Health, College of Health Science, QU Health, Qatar University, Doha 2713, Qatar; nislam@qu.edu.qa

**Keywords:** premarital screening program, knowledge, attitude, perception, health behavior, disease prevention, public health, Qatar

## Abstract

Premarital screening (PMS) is a primary preventive measure to decrease the incidence of certain genetic disorders and sexually transmitted diseases. This study aimed to explore the knowledge and perception of and the attitude toward PMS and predictors of knowledge and attitude. A cross-sectional study was conducted among Qatar University students using an online survey. Multivariable regression analyses were used to identify factors associated with PMS knowledge and attitude. A total of 476 students participated in the study; 424 (89.1%) were females; two-thirds were 18–21 years old. Only 100 participants had heard about PMS. Knowledge of PMS was significantly associated with females, students enrolled in a health-related college, and non-consanguineous marriage of a participant’s parents. The majority of the participants agreed that genetic diseases are psychological and economic burdens. For attitude, only 178 participants were willing to cancel marriages, given incompatible PMS results. The following factors were positively associated with attitude: PMS knowledge, enrollment in a health-related college, and the belief that PMS does not interfere with destiny. Our study findings revealed that despite the mandatory PMS in Qatar, the study participants, future couples, had low knowledge about the program. Therefore, strategies to increase awareness of PMS should be considered toward improving its outcomes.

## 1. Introduction

Premarital screening (PMS) is an efficient strategy for the primary prevention of specific genetic disorders and sexually transmitted diseases (STDs) [1]. PMS is a screening program offered to couples planning to get married in order to identify carriers of certain genetic diseases, e.g., sickle cell disease and thalassemia. These carriers are usually asymptomatic but can transmit such diseases to their future children if both couples are carriers. PMS is also used to test certain STDs, e.g., acquired immunodeficiency syndrome (AIDS) and hepatitis B and C, with the aim of reducing the incidence of genetic conditions and sexually transmitted diseases, hence minimizing the associated burden [2]. Partners with incompatible PMS results are usually offered counseling sessions so they can make informed decisions about their marriage, which might include marriage cancellation [3].

PMS has been implemented in several countries worldwide [2]. Cyprus was the first country that implemented mandatory PMS in 1973 before marriage, i.e., screening for β-thalassemia [4]. PMS was also mandated in other countries, especially those with high consanguinity rates, such as countries in the Middle East [5]. It was implemented in Iran in 1997, in Saudi Arabia and Jordan in 2004, in Bahrain in 2005, and in the United Arab Emirates in 2011 [2,6]. The diseases screened as part of PMS vary across countries [1]. For example, in Italy, screening is mandatory only for thalassemia [4], while in Saudi Arabia, it is mandatory for sickle cell disease, thalassemia, HIV, and hepatitis B and C [7]. In Egypt, PMS is mandatory only to conduct hemoglobinopathy screening [8].

In Qatar, PMS was initiated in 2009. It is a mandatory requirement for all Qatari planning to get married since its initiation and is offered in primary healthcare centers, while it was mandated for residents in 2012 and is offered in private clinics/hospitals. The test includes mandatory screening for thalassemia, sickle cell disease, classical homocystinuria, and cystic fibrosis, as well as STDs, including hepatitis B, hepatitis C, rubella, and HIV/AIDS, while optional screening is provided for spinal muscular atrophy (SMA) [2]. The turnaround time for PMS is 7 to 10 days but takes up to a month in case of undertaking the optional SMA test.

The need for mandating PMS in Qatar arose from the nature of increased consanguinity in the country, estimated at 54%, associated with the increased rates of several genetic disorders [9]. For example, Qatar has the highest incidence of classical homocystinuria globally (~1:1800) [9]. Moreover, there is an increased rate of hemoglobinopathies; the prevalence of β-thalassemia among Qataris is 30.4%, while the prevalence of α-thalassemia and sickle cell disease is estimated at 12.8% and 14.2%, respectively [10]. For STDs, the prevalence of hepatitis C is 0.51% [11], while there is a lack of published epidemiological data on the other STDs.

PMS and counseling have shown evidence of effectiveness in reducing the incidence of genetic diseases such as β-thalassemia as well as reduction in at-risk couples from getting married [12,13]. In their scoping review, Saffi and Howard (2015) showed that PMS reduced at-risk marriages by 2–58% and at-risk birth with β-thalassemia by 65–100% [6,12,13]. However, other factors play a role in PMS effectiveness.

Several studies have shown that after receiving the PMS results and counselling, the decision of couples depends on the couples’ knowledge and perceptions of the PMS program and consequences of its related diseases [14,15]. Therefore, the higher the level of knowledge, the lower the probability of having an at-risk marriage. Knowledge levels are affected by socio-demographical factors, such as age, educational status, and attendance of a medical or health-related college among others [14,15,16,17,18]. In Qatar, despite having a mandatory PMS program since 2009, there is a dearth of studies on the knowledge and attitudes toward the PMS program; only one recently published study explored PMS knowledge but among married individuals. The study [18] showed that knowledge levels were low and practical engagement in the program was only modest [18]. Further, the fact that couples have the right to get married regardless of their PMS results creates a challenge to the effectiveness of PMS in reducing rates of genetic diseases and STDs. Thus, it is important to understand the factors that play a role in PMS effectiveness, including marriage cancellation in the case of incompatible PMS results. It is imperative to investigate these factors, especially as perceived by future couples or the unmarried young population. Additionally, there is no study on the perception and knowledge of PMS among this segment of population in Qatar. Therefore, this study aimed to explore: (a) the level of knowledge regarding PMS; (b) the attitude toward PMS, defined as a predisposition to cancel marriage in the case of incompatible PMS results; and (c) factors associated with attitude among unmarried students or future couples.

## 2. Materials and Methods

### 2.1. Study Design

This cross-sectional study was conducted through an online survey.

### 2.2. Participants

The participants included Qatar University (QU), the largest national university, graduate and undergraduate, unmarried, ≥18-year-old students registered during spring 2020. Students were instructed to exit the survey if the inclusion criteria were not met, i.e., currently married and/or ˂18 years old. All participants who completed the survey were included. This study was approved by Qatar University’s Institutional Review Board (IRB; approval no. 1113-E/19). It was performed in concordance with QU-IRB regulations, which are concordant with the Helsinki Declaration of Ethical Principles for Medical Research. Participants provided informed consent electronically. Participants’ survey data were collected anonymously and stored safely on a laptop locked with a password. Focus group audio-recordings were transcribed and were stored securely using a research ID number only. Any identifying information was deleted before the file was stored; these de-identifiable data will be kept for 3 years and then destroyed. Only research staff have access to this information.

### 2.3. Data Collection

A non-probability, convenient sampling method was used to recruit the students. This method is suggested by researchers for using readily available subjects who volunteer to participate in a study [19]. Data were collected from 1 February 2020 to 15 March 2020 through a self-administered questionnaire. A link to the survey was sent to all QU students registered in spring 2020 via email with the help of the QU Institutional Research and Analytical Department. To increase the response rate, two email reminders were sent to the students at 2-week intervals. The questionnaire was available in Arabic and English. The total number of participants in the survey was 476 (roughly estimated response rate of 2.3%, no available data on married students to be excluded from the denominator).

### 2.4. Study Instrument

#### 2.4.1. Development of the Questionnaire

The questionnaire was developed through three phases.

Phase I: Item generation. Items were developed using two approaches: (a) literature review, where some questions were adopted [1,15,17,20,21], and (b) focus groups, with a 60 min focus group session comprising 8 members of the Qatar University Representative Board to establish a culturally responsive questionnaire. The questions for the focus groups were open ended, based on the Health Belief Model (see Appendix A) [22,23]. The focus group consisted of students who met the inclusion criteria of the study participants.

Phase II: Content validation and face validity of the questionnaire were established through expert panel reviews. The panel included two genetic counselors, two STD experts, and two members from the Social and Economic Survey Research Institute/Qatar University. They examined the degree to which the instrument could fully assess or measure the constructs of interest, e.g., knowledge of STDs and genetic diseases.

Phase III: Pilot testing: To establish the face validity of the questionnaire and its understandability, 20 students completed the questionnaire. Comments and feedback by respondents were considered; accordingly, three questions were deleted from the original questionnaire.

The questionnaire comprised three main parts: (a) a demographic part (15 questions); (b) knowledge about PMS and its related diseases (9 questions), and (c) perception of PMS (9 questions), in addition to and a question on attitude.

#### 2.4.2. Qualitative Analysis

First, the recorded focus group interviews were transcribed by one of the investigators (AR) and checked against the handwritten notes taken through the interviews. Then, the interviews were analyzed using content analysis. This involved identification of the most frequently repeated text through the application of a structured, systematic review from which themes can be derived from the content. The questionnaire was revised accordingly, e.g., added questions on religion interference with the PMS test.

Originally, all documents (focus group questions, questionnaire) were developed in English. However, Arabic versions were used to increase the response rate. The back-to-back translation was used by two bilingual expert translators in health science to check for the accuracy of the content [24].

### 2.5. Measures

There were two outcomes: attitude to cancel marriage in the case of incompatible PMS results (yes, no) and knowledge of PMS scores (sum of correct answers of questions 14, 15, 17, and 18; see Appendix A).

### 2.6. Statistical Analysis

Summary statistics were used to describe participants’ characteristics, i.e., frequency and percentages for categorical variables and means with standard deviations for continuous variables. Bivariate analyses were conducted for PMS knowledge by participants’ characteristics. The PMS knowledge scores (dependent variable) were not normally distributed, as shown by the Shapiro-Wilk test (*p*-value < 0.05). This was expected due to the difference in the students’ background; those with a health background are likely to have more PMS knowledge compared to others. Since the dependent variable was not normally distributed, non-parametric tests were used. These included the Mann-Whitney (for variables with 2 categories) or the Kruskal-Wallis (for variables with 3 or more categories) test. To determine predictors of PMS knowledge, linear regression analyses were conducted in 2 steps. First, a univariate regression analysis was conducted to examine the association between PMS knowledge and each of the independent sociodemographic variables (age, gender, nationality, marital status, income, college, and year of study; Table 1) as well as variables frequently reported in the literature to be associated with PMS knowledge, i.e., consanguinity marriage between a participant’s parents and having a family member with a genetic disease [16]. Second, a variable with a *p*-value of <0.25 [25] was entered in the multivariable linear regression model using a forward stepwise variable selection technique; variables were entered in the model one by one. A variable with a *p*-value of <0.05 was kept in the model; otherwise, it was removed, except those documented in the literature to be associated with PMS knowledge.

The same approach was used to determine predictors of attitude (cancellation of marriage with incompatible results (yes, no)) but via binary logistic regression analyses. The potential independent variables included socio-demographics, knowledge about PMS, genetic disease knowledge (sum of correct answers of questions 20 and 21), STD knowledge (sum of correct answers of questions 34 and 35), consanguinity between parents of a participant (yes, no), having a family member with genetic disease (yes, no), awareness of any Islamic fatwa (edict) approving abortion (yes, no), participants’ preference of consanguineous marriages (Likert scale of 5 points), and participants’ perception on a Likert scale of 5 points (1 = strongly agree, 5 = strongly disagree) of seven parameters. A *p*-value of <0.05 was considered statistically significant. Analyses were performed in SPSS v26 (IBM Corp, 2019. Armonk, NY, USA: IBM Corp).

## 3. Results

A culturally responsive questionnaire was developed based on the literature and a qualitative study. The questionnaire was examined for content and face validities. The final version comprised 34 questions with three main parts: (a) a demographic part (15 questions), (b) knowledge about PMS and its related diseases (9 questions), and (c) perception of PMS (9 questions), in addition to a question on attitude.

### 3.1. Participants’ Characteristics

A total of 476 students participated in the study. As shown in Table 1, the majority of the participants were females (89.1%), about two-thirds were in the age group of 18–21 years, less than half (44.5%) were Qataris, and 43.1% enrolled in health-related colleges. About one-fourth of the participants’ parents were relatives (first or second cousins), and almost one-fifth of the respondents had a family member diagnosed with a genetic disease.

### 3.2. Knowledge

Among respondents, 376 (79.0%) students had heard about PMS from academic courses (*n* = 88, 18.5%) or friends (*n* = 87, 18.3%), followed by social media (*n* = 57, 12%). The mean score of PMS knowledge was 2.81 ± 1.0 (out of 5 scores).

#### Factors Associated with PMS Knowledge

As shown in Table 2, gender, age group, nationality, college, consanguinity between participant’s parents, and having a family member with a genetic disease were significantly (*p* ˂ 0.05) associated with PMS knowledge based on the univariate analysis. However, the multivariable regression analysis (Table 3) showed that the female gender (vs. male; β = 0.13, *p* < 0.01), studying in a health-related college (vs. non-health college; β = 0.31, *p* < 0.001), students’ parents with more distant relations (β = 0.80, *p* = 0.002), and parents not related compared to first-cousin parents (β = 0.41, *p* = 0.04) were significantly associated with higher knowledge scores.

In contrast, a Qatari student (vs. non-Qatari; β = −0.51, *p*= ˂ 0.01), absence of genetic diseases among family members (vs. having a member with a genetic disease; β = −0.57, *p* ˂ 0.001), and students who had not heard about premarital screening (β = −1.55, *p* < 0.001) were significantly associated with lower knowledge scores.

### 3.3. Factors Associated with Attitude (Marriage Cancellation)

A total of 178 (37.4%) participants would cancel a marriage, given incompatible PMS results. Adjusted odds ratios of factors associated with attitude are presented in Table 4. Students in health-related colleges were 80.0% more likely to cancel the marriage compared to students in other colleges (odds ratio (OR) = 1.80, 95% confidence interval (CI) = 1.08–2.99). Further, students in their second, third, or fourth year of university study were almost twice as likely to cancel the marriage compared to those in their first year (Table 4). As PMS knowledge increased by one score, the probability of marriage cancellation increased by 24%. Students who perceived that PMS did not interfere with a belief in destiny were 90.0% more likely to cancel than those who believed (1.90; 95% CI = 1.17–3.11).

### 3.4. Perception

Participants’ perceptions of PMS are presented in Table 5. Most participants agreed, i.e., “strongly agree” and “somewhat agree,” that genetic diseases are psychological and economic burdens (*n* = 282, 80.0% and *n* = 260, 73.2%, respectively). Less than half of the participants (*n* = 153, 43.0%) perceived that genetic testing could lead to a denial of marriage. In addition, 38 (10.7%) participants agreed that “genetic testing will do more harm than good,” while 57 (16.0%) perceived that premarital screening interferes with a belief in destiny. One-third (*n* = 118, 33.3%) of the participants agreed that laws and regulations should ban marriage if there is a chance of having a child affected by a genetic disease.

## 4. Discussion

Premarital screening is an important primary prevention strategy, especially in highly consanguineous populations [26]. This is the first study in Qatar that assessed the knowledge and perception of and the attitude toward PMS among unmarried individuals. More importantly, the study revealed low PMS knowledge scores among university-level students, while the probability of at-risk marriage cancellation increased by increasing PMS knowledge.

### 4.1. PMS Knowledge

The level of knowledge among the participants in our study was low, with almost 56% of students having correct information about PMS and related diseases. This finding is consistent with findings from the region. In Saudi Arabia, a recent population-based study showed that only 9.2% (*n* = 575) participants had satisfactory knowledge, while 52.4% (*n* = 3283) participants had fair knowledge about PMS programs [27]. Another study in Oman reported a similar finding, i.e., half of the participants were unaware of premarital testing despite the majority (79%) having heard about PMS [28]. Indeed, these results underline the need for health education programs to increase awareness about PMS programs that aim to decrease the burden of some genetic and sexually transmitted diseases among future couples, especially in countries with high prevalence of consanguineous marriages and hereditary diseases.

Our results showed that females are more likely to have higher knowledge scores than males. Similar findings were reported in the literature from the Gulf region, Saudi Arabia, and Kuwait [29,30]. Another Nigerian study [31] demonstrated similar findings in female students compared to males. In Syria, university female students had more knowledge of and a positive attitude toward premarital genetic testing [32]. In contrast, adult males had more knowledge about premarital carrier screening compared to females in Oman [28]. However, no gender difference was observed in the study conducted by Bener et al. in Qatar [18]; this might be due to the difference in the target groups. Variation in the findings regarding difference in knowledge among genders could be attributed to cultural and socioeconomic factors. For example, culturally, females tend to be more knowledgeable about PMS and marriage-related matters since they are targeted by different education programs [33].

Further, students from health colleges had significantly higher PMS knowledge scores than students enrolled in non-health colleges. Similar findings have been reported in studies conducted in Saudi Arabia [33,34,35]. This observation is expected because students are more likely to learn about PMS and/or its related diseases as part of their academic curriculum. In a study that assessed and promoted premarital care services in Egypt, medical students showed a significant increase in the knowledge of and positive attitude toward premarital care services after undergoing educational sessions [36].

Compared to those who did not have a genetic disease, a participant or a family member with a genetic disease had higher PMS knowledge scores. Similar findings were reported by Kuwaiti [30] and Saudi [33] studies. This could be due to the resources available to families with affected individuals, such as educational materials and counseling sessions that widen their knowledge. Additionally, individuals with a family history of a certain genetic disorder are likely more interested in learning about the disease out of their concern of its possible impact on their own families or future offspring. In Qatar, for example, couples can get tested for known familial pathogenic variants as part of their PMS to clarify their carrier status. Thus, some couples planning to get married, especially in consanguineous unions, are promoted to seek information about PMS [37]. We can also anticipate that the perceived severity of the familial genetic condition can promote the couple to know more about the different aspects of the condition and its management, thus increasing their knowledge about PMS.

Participants with distantly related parents or unrelated parents had higher knowledge scores. Likewise, a Kuwaiti study showed that unrelated spouses (cousins or other) had higher knowledge about PMS [30]. This might be due to other factors, such as parents’ level of education, that likely improved the knowledge of their children [30]. In addition, participants who had never heard about PMS had significantly lower knowledge scores than participants who did. This is expected and calls for educational programs/awareness campaigns among students [33]. Such programs are especially needed in non-health-related colleges, which can be conducted through academic courses and social media platforms that ranked third as the source of PMS knowledge after academic courses and friends in our study.

### 4.2. Perception

Our study showed that 41.4% of the participants strongly agreed that genetic testing would do more good than harm to the public. A previous study in Qatar showed that 28% of the participants strongly believed that PMS would do good and prevent unexpected outcomes [18]. The difference in the proportions is likely due to the difference in the participants’ level of education in the two studies, i.e., 66.9% were below the university level in Bener et al.’s study [18]. Furthermore, a study conducted to assess the perception of premarital genetic testing in young Jordanian individuals indicated that around 90% of the participants had a positive perception of premarital genetic screening since they believed that screening would help in reducing the chance of having affected children, which might be financially and emotionally exhausting [38]. Interestingly, about one-fourth (24.8%) of the participants agreed that laws and regulations should ban marriage with a chance of having a child affected with a genetic disease. Likewise, a recent Omani study showed that nearly one-third of the participants (36%) agreed on the implementation of laws and regulations to prevent high-risk marriages [15]. However, in Qatar, while PMS is mandatory, it is the couples’ full choice to continue or cancel the marriage but after having a genetic counselling session educating the couples about the disease and burden if they have incompatible PMS results and signing a form indicating their understanding of the genetic counseling session provided. The banning of high-risk couples from getting married raises an important ethical question regarding one’s freedom to choose one’s partner. Furthermore, with the availability of other options to overcome the risk of having an affected child, such as preimplantation genetic testing and prenatal diagnosis, which are more available and accessible now, there is no legal demand to ban the marriage of at-risk couples.

### 4.3. Attitude

Only one-third (37.4%) of the participants were willing to cancel their marriage, given incompatible PMS results. In other studies, this percentage was higher. For example, it was 82.9% in a study at Taif University [39], 67.1% in another study in Saudi Arabia [33], and more than 60% in a population-based study in Saudi Arabia [40]. This could be due to the availability of prenatal diagnosis and pregnancy termination (under certain circumstances) in Qatar, as shown in other studies [41].

However, in our study, there were four predictors for marriage cancellation, as described above. With regard to PMS knowledge and enrollment in a health-related college, a study conducted in Saudi Arabia also showed a significant correlation between PMS knowledge and attitude [42]. This underlines the importance of increasing PMS knowledge, particularly for future couples to help them make informed decisions. Further, an Egyptian study found that medical students are significantly more willing to change their decisions about a PMS-incompatible marriage than non-medical students [14]. Similarly, students at higher levels of university study were more likely to cancel the marriage; this could be due to more exposure to PMS information, i.e., through university courses or other resources available in an academic environment.

Interestingly, our results demonstrated a significant association between the students’ willingness to cancel the marriage and the belief that PMS does not interfere with destiny. This could reflect the level of religious awareness among this high-level-educated population and the importance of dissemination of clear religious messages supporting PMS as a preventive strategy toward a healthy population. A study at Al-Taif University showed a different association, where only 5.2% of students disagreed that PMS interferes with destiny [39]. A similar misconception about Islamic principles that led to PMS rejection was also reported in other studies [1,43,44,45]. Therefore, it is essential to engage religious scholars in addressing such misconceptions among future couples to improve the effectiveness of PMS and reduce the burden of its associated diseases. In a study in Saudi Arabia [33], 87.6% of the participants agreed that religious scholars should be part of the PMS promotion through their public speech and discussions. Further, another Saudi study [43] showed that participation of religious scholars and the presence of a religious fatwa could positively change the attitude of about 53% of the study’s participants, who did not initially accept the concept of premarital health counseling. In Western countries, religious scholars also take part in premarital counseling [46,47].

### 4.4. Limitation

This study had some limitations, so our results should be interpreted with caution. First, the study had a low response rate of 2.3%; thus, our results could not be generalized. However, our results are coherent with a recent Qatari study in a different population, and it is likely that the generalizability issue is less serious [18]. The low response rate could be due to two major factors: (a) addressing a culturally sensitive topic, i.e., inherited diseases in a family, and (b) the COVID-19 crisis, which, at the beginning when the survey was conducted, disturbed individuals’ mental health and shifted priorities and daily life practices’ focus toward COVID-19 protection and survival. Second, the questionnaire was not validated due to limited resources; nevertheless, the questionnaire was reviewed for reliability by six experts in the field (two genetic counselors, two STDs experts, and two experienced researchers in survey studies). Third, the questionnaire was self-reported and could be subjected to social decrement, where individuals responded in a way to show their adherence to the ideal situation. This could have overestimated the attitude toward cancellation of marriage with incompatible PMS results; however, only one-third expressed their willingness to cancel a marriage with incompatible PMS results. Fourth, there could be a self-selection bias, which also might not accurately represent the general population. Finally, due to the nature of the design, i.e., cross-sectional, causal conclusions should not be drawn.

## 5. Conclusions

This is the first study to explore the knowledge and perception of and the attitude toward PMS in Qatar among unmarried participants (future couples). It showed a low knowledge level of PMS among participants, which is probably lower among the public, particularly those with low education levels. It is not sufficient to have a mandatory PMS program; increasing the awareness of PMS among the population is essential to decrease the rate of marriage with chances of having affected children. This finding is expected to aid decision makers when devising strategies on reducing the burden of genetic diseases and STDs associated with PMS. This could be done through conducting educational/awareness programs or public awareness campaigns with emphasis on PMS and genetic disease prevention. Additionally, increasing awareness about these diseases at the early stage of life through school curricula would be an effective strategy. Further, recruiting religious scholars to increase the knowledge levels of PMS and resolve any related misconceptions is essential for adopting healthy practices toward a healthy community. For future research, we recommend follow-up studies to assess the effectiveness of the implementation of mandatory PMS programs on the related diseases.

## Figures and Tables

**Table 1 ijerph-19-04418-t001:** Sociodemographic characteristics of the participants (N = 476).

Characteristics	*n* (%) *
**Gender**	
Female	424 (89.1)
Male	40 (8.4)
**Age group (in years)**	
18–21	325 (68.3)
22–25	104 (21.9)
26–29	20 (4.2)
≥30	15 (3.2)
**Nationality**	
Qatari	212 (44.5)
Non-Qatari	251 (52.7)
**Religious status**	
Islam	459 (96.4)
Others	6 (1.0)
**College**	
College of Business & Economics	29 (6.1)
College of Sharia	59 (12.4)
College of Law	31 (6.5)
College of Engineering	76 (16.0)
College of Arts & Sciences	33 (6.9)
College of Health Sciences	123 (25.9)
College of Medicine	48 (10.1)
College of Education	30 (6.3)
College of Pharmacy	34 (7.1)
**Year of study in college**	
1st Year	109 (22.9)
2nd Year	144 (30.3)
3rd Year	84 (17.7)
4th Year	73 (15.3)
5th Year or more	51 (10.7)
**Marital status**	
Single	414 (87.0)
Engaged	32 (6.7)
Previously married **	12 (2.6)
**Monthly household income (excluding drivers’ and maids’ salaries)**	
Less than QAR 11,000 †	100 (21.0)
QAR 11,000–19,999	126 (26.5)
QAR 20,000 or more	222 (46.6)
**Relation between participants’ parents**	
First cousins	75 (15.8)
Second cousins	48 (10.1)
More distant relation	76 (16.0)
Not related at all	263 (55.3)
**Family diagnosed with genetic disease**	
Yes	103 (21.6)
No	258 (54.2)
Do not know	100 (21.0)

* Subgroups that do not add to 100% indicate missing data; ** previously married included separated, divorced, and widowed; † equivalent to USD 3021.15, i.e., 1 QAR = USD 0.27 for year 2020.

**Table 2 ijerph-19-04418-t002:** Association between knowledge and participants’ characteristics (univariate analysis results).

Characteristics	Knowledge Scores on PMS (Mean ± SD)/5	*p*-Value
**Gender**		0.03 ^a^
Male	2.33 ± 1.79	
Female	2.94 ± 1.65	
**Age group (in years)**		0.03 ^b^
18–21	2.94 ± 1.65	
22–25	2.53 ± 1.73	
26–29	3.55 ± 1.50	
≥30	3.27 ± 1.58	
**Nationality**		<0.01 ^a^
Qatari	2.55 ± 1.65	
Non-Qatari	3.16 ± 1.64	
**College**		<0.01 ^b^
College of Business & Economics	1.76 ± 1.64	
College of Sharia	2.61 ± 1.55	
College of Law	2.29 ± 1.70	
College of Engineering	2.22 ± 1.55	
College of Arts & Sciences	2.73 ± 1.64	
College of Health Sciences	3.70 ± 1.52	
College of Medicine	3.44 ± 1.50	
College of Education	2.70 ± 1.34	
College of Pharmacy	3.00 ± 1.69	
**Year of study**		0.34 ^b^
1st Year	2.82 ± 1.65	
2nd Year	3.08 ± 1.58	
3rd Year	2.76 ± 1.75	
4th Year	2.96 ± 1.66	
5th Year or more	2.84 ± 1.83	
Marital status		0.47 ^a^
Single (never married)	2.91 ± 1.67	
Engaged	2.56 ± 1.79	
**Relation between parents**		<0.01 ^b^
First cousins	2.48 ± 1.61	
Second cousins	2.33 ± 1.78	
More distant relation	3.29 ± 1.47	
Not related at all	3.00 ± 1.68	
**Family member diagnosed with a genetic disease**		<0.01 ^a^
Yes	3.37 ± 1.55	
No	2.79 ± 1.70	
Do not know	2.64 ± 1.62	
**Have you ever undergone premarital screening?**		0.35 ^a^
Yes	2.64 ± 1.75	
No	2.91 ± 1.67	

^a^ Mann-Whitney test; ^b^ Krukal-Wallis test.

**Table 3 ijerph-19-04418-t003:** Factors associated with PMS knowledge score (multivariable regression analysis).

Variable	β Coefficient	95% CI	*p*-Value
**Gender**			
Male	*Reference*		
Female	0.13	(0.12–1.30)	˂0.01
**Age (in years)**			
18–21	*Reference*		
22–25	−0.10	(−0.51–0.30)	0.62
26–29	0.36	(−0.39–1.12)	0.35
Older than 30	0.39	(−0.48–1.27)	0.38
**Nationality**			
Non-Qatari	*Reference*		
Qatari	−0.51	(−0.84–−0.18)	˂0.01
**College**			
Other colleges	*Reference*		
Health-related	0.31	(0.72–1.35)	˂0.001
**Year of study**			
1st Year	*Reference*		
2nd Year	0.18	(−0.19–0.56)	0.34
3rd Year	−0.12	(−0.56–0.32)	0.58
4th Year	0.32	(−0.18–0.83)	0.21
5th Year or more	0.32	(−0.29–0.93)	0.31
**Marital status**			
Engaged	*Reference*		
Single	0.39	(−0.18–0.97)	0.18
**Relation between parents**			
First cousins	*Reference*		
Second cousins	−0.12	(−0.66–0.42)	0.80
More distant relation	0.80	(0.30–1.28)	˂0.01
Not related at all	0.41	(0.01–0.89)	0.04
**Family member diagnosed with genetic disease**			
Yes	*Reference*		
No	−0.57	(−0.92–−0.22)	˂0.001
Do not know	−0.49	(−0.91–−0.07)	0.02
**Have you heard about premarital screening or genetic testing?**			
Yes	*Reference*		
No	−1.55	(−1.91–1.19)	˂0.001

**Table 4 ijerph-19-04418-t004:** Adjusted odds ratio (OR) of attitude (marriage cancellation in the case of PMS incompatible results) by participants’ characteristics.

Variable	Adjusted OR	95% CI	*p*-Value
**Nationality**			
Non-Qatari	*Reference*		
Qatari	1.50	(0.93–2.41)	0.95
**College**			
Other colleges	*Reference*		
Health-related-colleges	1.80	(1.08–2.99)	0.02
**Year of study**			
1st Year	*Reference*		
2nd Year	2.65	(1.40–5.02)	0.003
3rd Year	2.35	(1.16–4.77)	0.02
4th Year	2.50	(1.12–5.59)	0.03
5th Year or more	2.01	(0.74–5.49)	0.17
**Knowledge scores of PMS**	1.24	(1.06–1.46)	0.009
**Premarital screening does not interfere with a belief in destiny.**			
Disagree	*Reference*		
Agree	1.90	(1.17–3.11)	0.01

PMS, premarital screening; CI, confidence interval. A forward stepwise variable selection model was used for identifying significant predictors. For the total list of variables examined, see the description in the Statistical Analysis section.

**Table 5 ijerph-19-04418-t005:** Participants’ perception of PMS.

	What Is Your Extent of Agreement with Each of the Below Statements? *n* (%)					
Statement	Strongly Agree	Somewhat Agree	Neutral	Disagree	Strongly Disagree	Mean Score * (SD)
Genetic diseases can have a psychological burden on families (*n* = 357).	149 (41.7)	133 (37.3)	53 (14.8)	22 (6.2)	0 (0.0)	4.15 (0.89)
Genetic diseases can have an economic burden on families (*n* = 355).	133 (37.5)	127 (35.8)	51 (14.4)	20 (5.6)	24 (6.8)	3.92 (1.16)
Genetic testing could lead to denial of marriage for a couple (*n* = 356).	110 (30.9)	143 (40.2)	68 (19.1)	33 (9.3)	2 (0.6)	3.92 (0.96)
Genetic testing will do more harm than good for society (*n* = 355).	9 (2.5)	29 (8.2)	40 (11.3)	130 (36.6)	147 (41.4)	1.94 (1.04)
Premarital screening does not interfere with a belief in destiny (*n* = 355).	147 (41.4)	84 (23.7)	67 (18.9)	44 (12.4)	13 (3.7)	3.87 (1.19)
Screening for sexually transmitted diseases is necessary before marriage (*n* = 341).	270 (79.2)	45 (13.2)	2 (0.6)	21 (6.2)	3 (0.9)	4.64 (0.85)
Law and regulation should ban the marriage if there is a chance of having a child affected with a genetic disease (*n* = 354).	40 (11.3)	78 (22.0)	50 (14.1)	94 (26.6)	92 (26.0)	2.66 (1.37)

* Mean score of individual perception items, out of a score of 5, on a 1–5 Likert scale (5 = strongly agree, 4 = somewhat agree, 3 = neutral, 2 = disagree, 1 = strongly disagree).

## Data Availability

The datasets used in this study are available from the corresponding authors on reasonable request.

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
