# Peer review of "Knowledge and Perception of and Attitude toward a Premarital Screening Program in Qatar: A Cross-Sectional Study"

_ijerph, 2022, doi:10.3390/ijerph19074418_

Round 1

Reviewer 1 Report

Authors made considerable recommendations for ameliorating their mscr. I have no further concern about the quality of the paper.

Author Response

Reviewer-1

Authors made considerable recommendations for ameliorating their mscr. I have no further concern about the quality of the paper.

Authors:

Thank you for taking the time to read our manuscript and for valuable comments.

Reviewer 2 Report

This study demonstrate, basically, descriptive statistics only. While the authors claim that they conducted univariate/multivariate regression analyses, it is expected that they analyze the issue from causal-inference viewpoint. Otherwise, there will be no informative policy suggestions on what to do with the PMS. It is also a significant disadvantage, as the authors understand, that the response rate remains quite low, which makes even descriptive statistics misleading.

Author Response

Reviewer: 2

This study demonstrate, basically, descriptive statistics only. While the authors claim that they conducted univariate/multivariate regression analyses, it is expected that they analyze the issue from causal-inference viewpoint. Otherwise, there will be no informative policy suggestions on what to do with the PMS. It is also a significant disadvantage, as the authors understand, that the response rate remains quite low, which makes even descriptive statistics misleading.

Authors:

As this is a cross-sectional study, we agree that causal conclusions should not be drawn. We made it clear in the text under limitations, lines 386-7.

Reviewer 3 Report

Please, check the references

Please,  explain better the limitation - study had a low response rate of 2.3%

Author Response

Reviewer: 3

Please, check the references

Authors:

Checked and edited

Reviewer: 3

Please, explain better the limitation - study had a low response rate of 2.3%

Authors:

Low response rate was explained, please see lines 373-377.

Round 2

Reviewer 2 Report

Unfortunately, the authors failed to overcome the concerns pointed out by the reviewer.

Author Response

Thank you for taking the time to review our paper.

This manuscript is a resubmission of an earlier submission. The following is a list of the peer review reports and author responses from that submission.

Round 1

Reviewer 1 Report

Authors discuss a  subject that is equally  sensitive  and important for islamic or african communities.

Although the questionnaire is poor and , it is obvious that such societies could not manage more or extended questions. The evaluation of the questionnaire is of high level. The only amelioration they could  make are:

  1. to compare  the observed levels of knowledge with other societies
  2. to further discuss measures of health education or mitigation.

discuss

More importantly, though the questionnaire was not previously validated, I strongly recommend the publication of the article  after adding the info I suggested in the previous paragraph. The reason is that such works would help practitioners as well as societies to advance public health surveillance measures albeit religious misfunctionalities.

Author Response

Dear Prof. Dr. Paul B. Tchounwou,

Editor-in-Chief, International Journal of Environmental Research and Public Health

Dear Prof. Dr. Sara Rubinelli, Dr. Nicola Diviani and Dr. Claudia Zanini Guest Editors, Special Issue "Behavior Change in Health Promotion and Prevention of Chronic Diseases"

Re:  Knowledge, Perception and Attitude towards Premarital Screening Program in Qatar: A cross-sectional study (ijerph-1488186).

Many thanks for giving us the opportunity to respond to the reviewer’s comments. Our specific responses are detailed below.

Reviewer 1

Comments and Suggestions for Authors

Authors discuss a  subject that is equally  sensitive  and important for islamic or african communities.

Authors: Thank you for reading our paper and for your valuable comments.

Although the questionnaire is poor and, it is obvious that such societies could not manage more or extended questions. The evaluation of the questionnaire is of high level. The only amelioration they could make are:

  1. to compare  the observed levels of knowledge with other societies

Authors: Thank you for your comments. We have modified based on this suggestion in the manuscript (paragraph 2, line: 252-262).

  1. to further discuss measures of health education or mitigation.

Authors: We have modified based on this suggestion in the manuscript (paragraph 3, line: 301-304), (paragraph 4, line: 319-328), (paragraph 3, line: 348-350 and 353-355).

More importantly, though the questionnaire was not previously validated, I strongly recommend the publication of the article after adding the info I suggested in the previous paragraph. The reason is that such works would help practitioners as well as societies to advance public health surveillance measures albeit religious misfunctionalities.

Authors: Thank you for your comments. They added to the quality of the manuscript.

Reviewer 2 Report

I read your results well. Thanks for broadening my scope. Good luck to you.

Let me give you some comments. It is hoped that this will be reflected in high-quality research.Introduction
Introduction is very poor. Please refer to the comments below to make up for more.
1. Add a description of the PMS, such as the target and timing of the PMS program implemented in Qatar.
2. Add a description of the effectiveness and expectations of the PMS program implemented outside Qatar.
3. Describe the current situation of sexually transmitted diseases and hereditary diseases in Qatar.
4. Describe the need for research by referring to previous studies on similar topics.

Materials and Methods
1. Separate sections for ethical considerations in research.
2. Explain the rationale for setting the number of participants (ex. G-power)
3. Describe non-face-to-face collection in detail (response rate, etc.)
4. Write a more detailed focus group interview.
5. It is said that knowledge is not normally distributed. Describe the reason in detail and write the rationale.
6. What non-parametric tests did you do?

Results
1. List the results as described in Statistical Analysis.
2. Also describe the results of developing the questionnaire.

  Discussion
1. Add the ethical issues caused by the PMS program.
2. Also add a description of the expected effect of the study.
3. Add various comparisons with previous studies for each section.

Author Response

Dear Prof. Dr. Paul B. Tchounwou,

Editor-in-Chief, International Journal of Environmental Research and Public Health

Dear Prof. Dr. Sara Rubinelli, Dr. Nicola Diviani and Dr. Claudia Zanini Guest Editors, Special Issue "Behavior Change in Health Promotion and Prevention of Chronic Diseases"

Re:  Knowledge, Perception and Attitude towards Premarital Screening Program in Qatar: A cross-sectional study (ijerph-1488186).

Many thanks for giving us the opportunity to respond to the reviewer’s comments. Our specific responses are detailed below.

Reviewer 2

Comments and Suggestions for Authors

I read your results well. Thanks for broadening my scope. Good luck to you.

Authors: Thank you for your valuable comments.

Let me give you some comments. It is hoped that this will be reflected in high-quality research.

Introduction: Introduction is very poor. Please refer to the comments below to make up for more.
1. Add a description of the PMS, such as the target and timing of the PMS program implemented in Qatar.

Authors: We have added this in the introduction (paragraph 1, line: 47-53).

  1. Add a description of the effectiveness and expectations of the PMS program implemented outside Qatar.

Authors: We have added this in the introduction (paragraph-3, line: 64-67).

  1. Describe the current situation of sexually transmitted diseases and hereditary diseases in Qatar.

Authors: As requested, information has been added (paragraph 2, line: 55-61).

  1. Describe the need for research by referring to previous studies on similar topics.

Authors: We have added the requested information in the last paragraph of the introduction (paragraph-04, line: 70-84).

Materials and Methods
1. Separate sections for ethical considerations in research.

We have added a separate section in the manuscript (paragraph-01, line: 90-93).

  1. Explain the rationale for setting the number of participants (ex. G-power)

We used a convenient sampling method to recruit the students due the cultural sensitivity to genetic and STDs diseases. We have added this information in the methods (paragraph-01, line: 103-105).The limitation of this method with the low response rate was already stated under the “Limitation” part.

  1. Describe non-face-to-face collection in detail (response rate, etc.)

Authors: We have added the information under “Data Collection” (paragraph-01, line: 105-111).

  1. Write a more detailed focus group interview.

Authors: Supplementary S2 file on the interview questions has been added and more information was integrated under “Qualitative analysis” (paragraph-07, line: 141-144).

  1. It is said that knowledge is not normally distributed. Describe the reason in detail and write the rationale.

Authors: We have edited the methods section and explanation has been added (paragraph-01, line: 155).

  1. What non-parametric tests did you do?

Authors: They have been mentioned already, apologies for being unclear. Now we have edited the text under methods to become clear (paragraph-01, line: 158-159).

Results
1. List the results as described in Statistical Analysis.

Authors: results have been re-organized as described in the statistical analysis

  1. Also describe the results of developing the questionnaire.

Authors: We have added this in the results (paragraph-01, line: 186-190).

  Discussion
1. Add the ethical issues caused by the PMS program.

Authors: We have added this in the discussion (paragraph-03, line: 312-317).

  1. Also add a description of the expected effect of the study.

Authors: We have added this in the conclusion (paragraph-01, line: 383-388).

  1. Add various comparisons with previous studies for each section.

Authors: We have expanded the discussion by inserting more studies across the sections.

Round 2

Reviewer 2 Report

Thanks for the reply. But there are still doubts.
You must provide me with the final version. I cannot read the paper properly because the review item is turned on.
At the end of the introduction, you should explain why your research is necessary.
Why is the number of participants in Table 1 different for each item?
Move ethical considerations last.
Describe subject protection on sensitive matters.